# The Cascade of High-Voltage Pulsed Current Sources

Weigang Dong [1], Lei Chen [1], Jian Qiu [1,*], Haozheng Shi [2], Hui Zhao [1] and Kefu Liu [1,*]

[1] Department of Light Sources and Illuminating Engineering, School of Information Science and Technology, Fudan University, Shanghai 200433, China; dwg@fudan.edu.cn (W.D.); 21210720061@m.fudan.edu.cn (L.C.); hui_zhao@fudan.edu.cn (H.Z.)

[2] Shanghai Fudan Microelectronics Group Co., Ltd., 127 Guotai Road, Shanghai 200433, China; shihaozheng@msn.cn

[*] Correspondence: jqiu@fudan.edu.cn (J.Q.); kfliu@fudan.edu.cn (K.L.)

**Abstract:** Currently, pulsed adders are used as pulsed voltage sources maturely. However, their use as pulsed current sources is significantly limited due to circuit impedance and the characteristics of power devices. This paper presents a simple yet effective design for a pulsed current source, incorporating a solid-state Marx pulsed adder as the primary power source and an inductor for energy storage. In the pulsed current source, a Marx pulsed adder produces high voltage to charge the inductor. Then, the stored inductance energy is converted to generate current pulses on the load; the amplitude of the pulsed current is unaffected by the load impedance within a certain range. The pulsed current source can be designed as a standard module, and several modules can form a cascade system for producing current pulses with higher voltage. Finally, a pulsed current source was developed, which can produce adjustable current pulses with high voltage. The design principles, control methods and the effects of the distribution parameters are described. The feasibility of the cascade pulsed power system was validated in experiments. Nine modules were connected to generate pulses of current 10 A on a 15 kΩ resistor.

**Keywords:** cascade; pulsed current source; waveform analysis





## 1. Introduction

Pulsed power technology has been widely used in industrial manufacturing, environmental engineering, biological medicine, national defense and other fields [1–5]. A pulsed power system typically consists of three components: a primary power source, an intermediate power storage and a pulse forming network (PFN). In recent years, semiconductor devices have become the preferred power devices in pulsed power systems due to their advantages of high sensitivity, fast response speed, compact size and extended operational lifetime [6].

With the application of pulsed power technology expanding to broader fields, such as driver of accelerator [7], surface treatment of materials [8], ion source [9] and dielectric barrier discharge (DBD) [10,11], the applications mentioned above demand that the pulsed power system generate high voltage and stable current pulses under loads with different or varying impedance. In these cases, the traditional pulsed power generators with capacitive energy storage (CES) have great limitations because they are essentially voltage sources and their output current can be changed significantly by loads with different resistance. Pulsed current generators using inductive energy storage (IES) can satisfy this demand, and there have been many studies on inductive pulsed current generators [12–15]. When the current flowing through the inductor changes, counter electromotive force will be generated at both ends of the inductor to maintain the original current amplitude. In other words, this counter electromotive force does not allow the current in the inductor to change. In addition, current sources have better robustness than voltage sources in certain situations, such as driving a low-impedance load or a load that is prone to be short-circuited because there is no need to worry about overcurrent.

The main difficulty in the application of IES technology is to cut off the DC current during the working process. Therefore, the research on commutation technology has developed rapidly in recent years. According to the power electronic devices and commutation principle, commutation technology can be divided into the direct cutoff method and forced current zero formation [16]. As the key to commutation technology, the switches used in IES devices need excellent performance, like superior high speed, high voltage and high capacity, which are usually expensive and difficult to use [17].

In [18], a power supply system for a vacuum-arc ion source with a millisecond pulse length was proposed. The power supply system for the vacuum-arc ion source was intended to generate pulses with a length up to 2000 μs at an accelerating voltage up to 50 kV. The pulsed current was provided using the proposed IGBT-based power supply system operating in the burst mode. One important advantage of this system was a small amount of the stored energy transferred in load upon the breakdown in the accelerating gap. The disadvantages of this system included a strong dependence of the accelerating voltage amplitude on the load resistance, which complicated the control for the acceleration ion energy. The current pulse amplitude could reach 1 kA, but the stability of the pulsed current was not very good.

In [19], the proposed compact Marx Generator has achieved an output pulse with a rise time of 17 ns, a full width half maximum of 540 ns and an adjustable peak current of the pulse between 10 A and 3 kA into a 60 $\Omega$ $CuSO_4$ resistive load. The data collected from the testing operation in the $CuSO_4$ load show a good output waveform of standard. The Marx Generator is charged by a +70 kV high-voltage DC source. The current waveform does not have a platform, and a pneumatic-drive trigger switch is used, which would limit the operating frequency of the system.

In [20], a pulse constant current source with negative feedback of base current and voltage is designed. The experiment shows that the selection of appropriate devices can meet the requirements of an airbag ignition test, the output characteristics of the constant current source circuit can meet the requirements of 0–10 A adjustable and the frequency 0–5 kHz adjustable, and the establishment time is less than 50 μs. The method mentioned in [20] is usually used in lower-voltage situations, and the efficiency is not very high.

Currently, the solid-state Marx pulsed adder stands as the primary equipment for high-voltage pulsed power applications. It facilitates seamless circuit conversion by controlling the switching of multiple units [21–23]. In [24], a high-gain pulse generator topology is proposed that is suitable for plasma jet applications. The proposed topology utilizes an inductor as the isolation device for the solid-state Marx circuit. With the addition of only one switch, the inductor is used to charge and discharge, resulting in a high gain of output pulse amplitude. In [25], a developed device can generate pulses from 100 ns to 100 μs, up to 6.5 kV and 65 A with a maximum repetition rate of 5 MHz, from four independent and synchronized Marx modulators. Almost all pulsed generators of this type have voltage source characteristics; the output current cannot automatically adapt to the change in the load impedance, so they cannot meet some special application requirements.

Building on the capability of Marx pulsed adders, a novel high-voltage pulsed current source can be developed by integrating the Marx pulsed adder with inductive energy storage and enhancing the commutation circuit. The inductive current is precisely regulated through the charging voltage and time, both of which are governed by the Marx pulsed adder. It is essential for the circuit design to offer multiple current pathways, enabling effective energy transfers between the Marx pulsed adder, the inductor and the loads.

In this paper, an inductive pulsed current source is designed and manufactured as a standard module, which can generate square current pulses with high voltage. Through linking these modules to form a cascade system, relatively stable pulsed current pulses can be generated with a higher voltage output. When the insulation breakdown phenomenon occurs on the load, the cascade system can still work properly. Meanwhile, the cascade system can limit the output pulsed current to avoid damage to the load electrodes. Finally, the cascade system is used in energetic particle accelerators.

## 2. Pulsed Current Source Module

Figure 1 shows the circuit structure of the pulsed current source based on the solid-state Marx pulsed adder. The symbol i represents the sequence of each unit in the module. The switches in the same position of each unit are synchronous. Switches $S_{a-i}$, $S_{b-i}$ and $S_{c-i}$ are insulated gate bipolar transistors (IGBTs). Capacitor $C_i$ is the energy storage capacitor in the unit, and resistor R is the series equivalent resistance of the energy storage capacitors. $D_i$, $D_{1-i}$ and $D_{2-i}$ are fast recovery diodes. A resistive load, a silicon stack and an inductor are also included. The solid-state switches are driven by the passive method [26]. The driving circuit and the main circuit are isolated by magnetic cores [27]. The circuit operating process has been described in [28]. The module of the pulsed current source that can generate pulses of voltage 20 kV and current 10 A contains 24 units, and each energy storage capacitor in the unit should be charged to ~1000 V.

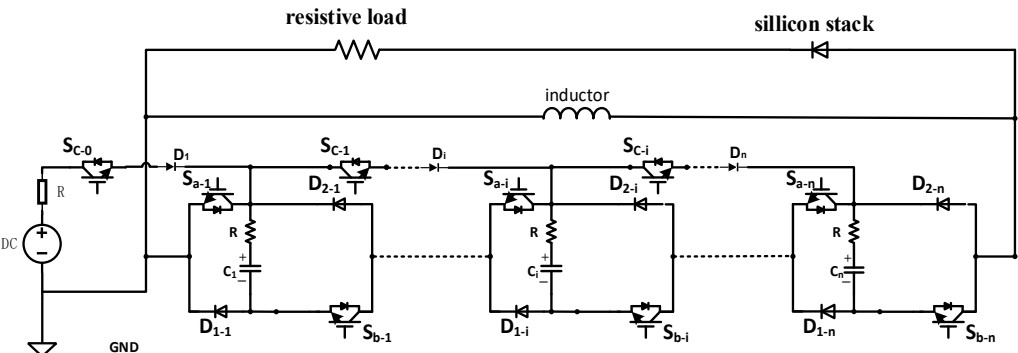

**Figure 1.** Circuit structure of the pulsed current source.

The pulsed current module has four operating processes: capacitance charging, inductance charging, load discharging and load current chopping. When working in repeated pulse mode, the pulsed current source has three circular operating processes after initialization: load current chopping synchronized with capacitance charging, inductance charging and load discharging. The initialization is similar to the single pulse mode. The detailed circuit analysis was shown in [28]. After that, switches $S_{b-i}$ and $S_{c-i}$ turn on, and switches $S_{a-i}$ keep off; the current source enters load current chopping and capacitance charging simultaneously. The procedure will last for a certain period during this stage to ensure that the capacitive voltage can reach the expected value again. Then, the source will come to the inductance charging stage and the load discharging stage in turn as the same as that in the single pulse mode. These three operating processes work in cycles so that the source outputs repeated pulses. Unlike the single pulse mode, the current of the inductor does not need to drop to zero in repeated pulse mode, which makes the time of load current chopping much shorter. However, under the repeated pulse mode, it is necessary to have an appropriate working time for each stage so that they can cooperate with each other to ensure the ideal repeated pulse output. This demands high precision and complexity from the control of this circuit. Figure 2 shows the theoretical waveforms of control signals, capacitance voltage and inductor current in repeated pulse mode. $I_{load}$ is the current passing the resistive load. $T_1$, $T_2$ and $T_3$ represent three adjacent periods of repeated pulse output.

Figure 3 shows the typical output waveform of the pulsed current source [28]. The solid-state Marx pulsed adder outputs a high voltage to charge the inductor; the initial output voltage of the Marx pulsed adder is approximately 22 kV. After 180 μs, the inductor current reaches approximately 10 A, and then the inductor discharges the resistance load of 500 Ω for 20 μs. From the experimental results, if keeping the same inductive charging time, the output voltage of the solid-state Marx pulsed adder can also change the inductive charging current. With the fixed capacity of the Marx adder, the lower the initial capacitor voltage, the smaller the energy on the storage capacitor; therefore, to charge the inductor to

the same energy (current), the more the total capacitance voltage decreases. In summary, the initial output voltage of the Marx pulsed adder shall be maintained at 24 kV if the inductance current is 10 A with a 2 kΩ load. Figure 4 is the photo of the pulsed current module that includes a Marx pulsed adder; each module contains 26 basic units, and the value of the DC voltage source should be approximately 800~1000 V for 20~24 kV pulse output.

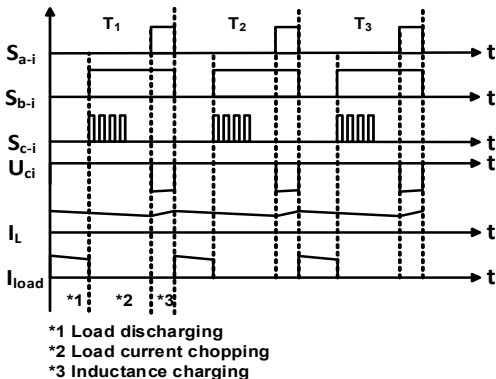

**Figure 2.** Theoretical waveforms in repeated pulse mode.

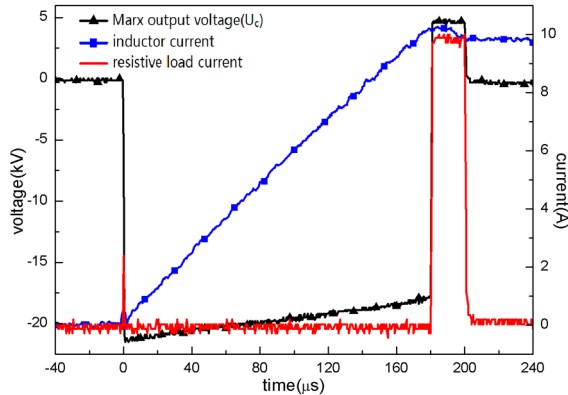

**Figure 3.** Typical waveform of the pulsed current source.

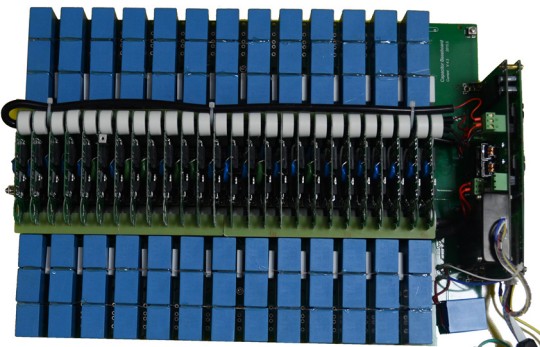

**Figure 4.** The photo of the packet cascade mode module.

## 3. Modular Cascade System

The pulsed current source based on the solid-state Marx pulsed adder can also obtain a higher output feature through the cascade. There are two ways of module cascade: overall cascade mode and packet cascade mode. The overall cascade mode refers to taking a complete pulse current source as the basic unit of the cascade, forming a larger power supply system through the connection between the pulse current sources. The

packet cascade refers to the grouping of inductors and solid Marx pulsed adder, cascading respectively, and then connecting the two groups to achieve a larger power system. The overall cascade mode is simpler and closer to the modular goal, but there are some problems in the actual system. The packet cascade, although more complex, is more adaptable to the differences between modules.

### 3.1. Overall Cascade Mode

The overall cascade mode module is shown in Figure 5. The module is integrated with the energy storage inductance. The cascade system is connected to the load and the silicon stack.

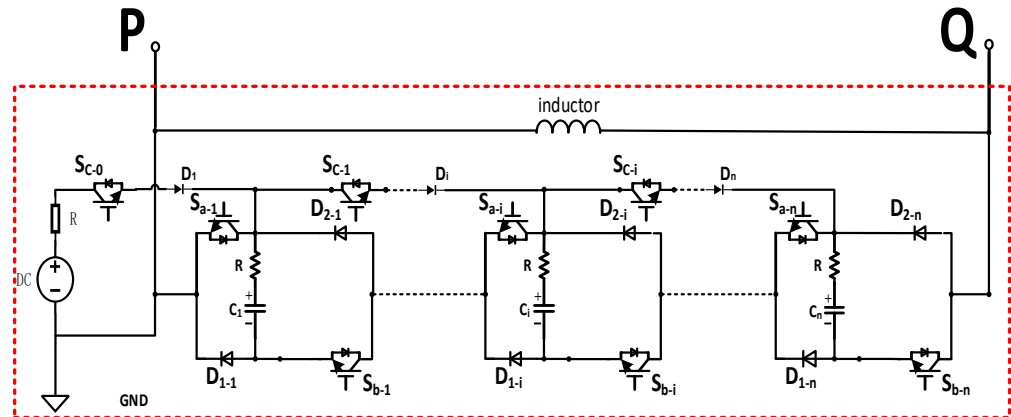

**Figure 5.** Overall cascade mode module.

Figure 6 shows the schematic diagram of the m-stage pulse current source system connection using the overall cascade mode. The low electric potential stage module Q port is connected to the upper electric potential stage P port, the 1 stage (lowest electric potential stage) P port is the system connection site, and the m stage (highest electric potential stage) Q port is connected to the series circuit composed of the load and silicon stack.

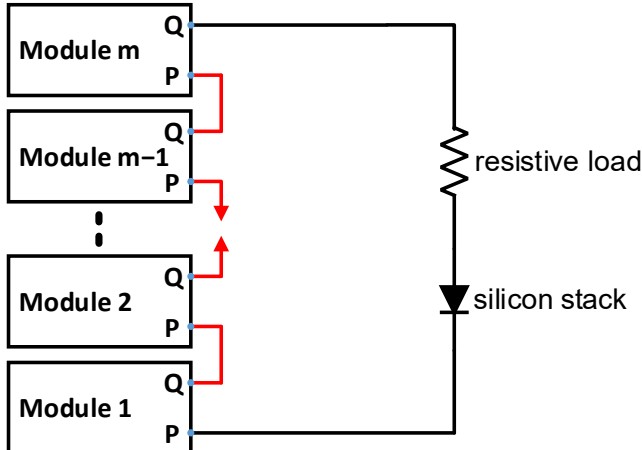

**Figure 6.** An m-stage pulse current source system in an overall cascade mode.

### 3.2. Packet Cascade Mode

The packet cascade mode module is shown in Figure 7. The module can be roughly regarded as a solid-state Marx pulsed adder. Compared with the overall cascade mode module, the energy storage inductance is removed from the modules, and they are separately connected outside the cascade units.

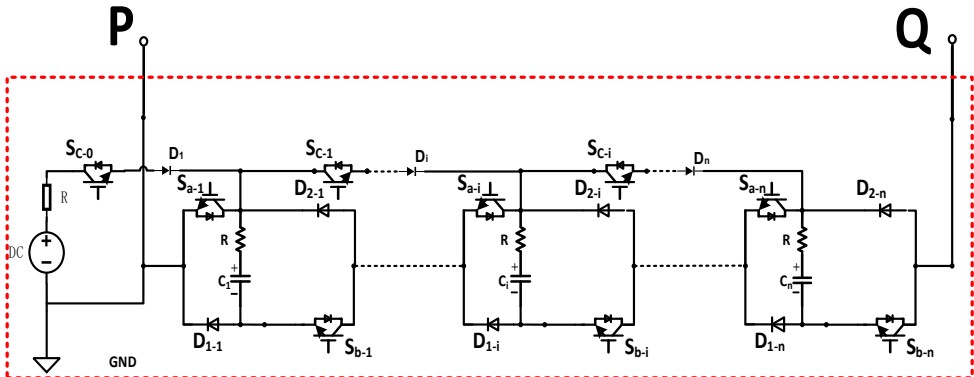

**Figure 7.** Packet cascade mode module.

Figure 8 shows the schematic diagram of the m-stage pulse current source system connected by the packet cascade mode. The solid-state Marx pulsed adder and the high-voltage inductance are divided into two parts. The number of modules n and the number of inductors k cannot be equal. The Q port of the lower electric potential stage module is connected to the upper stage P port, and the first stage 1 (lowest electric potential stage) P port is the system connection site, which connects the series inductor part and the series circuit composed of the load and silicon stack.

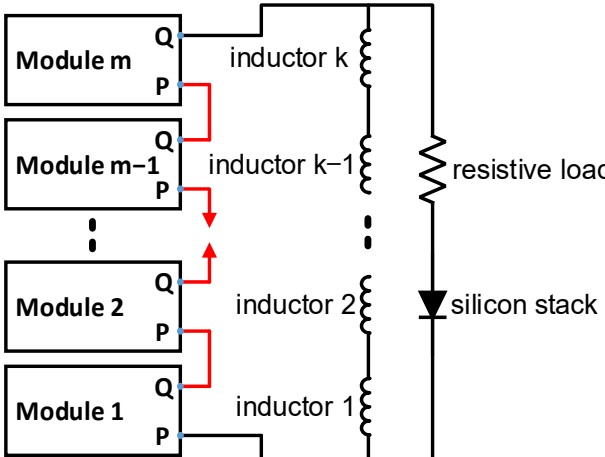

**Figure 8.** An n-stage pulse current source system using a packet cascade mode.

The composition of the packet cascade system requires the circuit of the solid-state Marx pulsed adder and inductors. The disassembly and maintenance are more complex than the overall cascade mode, but the packet cascade mode also has obvious advantages: the cascade inductance can be easily adjusted to accommodate different load and output requirements.

## 4. Discussion of the Cascade Approach

Although the pulse current source module ensures maximum consistency during the manufacturing process and the driving circuit is used to ensure that all switches are synchronized in strict accordance with the time sequence, it will still face some inevitable differences in the cascade working process. The solid-state Marx pulsed adder may have a different output voltage amplitude due to subtle differences in the charging voltage of the energy storage capacitor and damage to a certain Marx stack. Different amounts of energy storage and equivalent series resistance will also cause uneven inductor charging current. The cascade scheme must be able to ensure that the system still works properly when there are differences in module consistency.

According to the overall cascade circuits shown in Figures 4 and 5, in order to ensure the consistent current on the energy storage inductance of each pulsed current module, we need to control each stage separately to balance the differences between the modules by adjusting the inductor charging time and charging voltage. If there is no way to control each module, the overall cascade scheme is extremely sensitive to different inductive charging currents. The current output by the current source to the load will be the minimum inductive current, and the current on the load will rise slowly during the expected flat-top stage. If a certain stage of the Marx stack cannot output voltage and the inductive current on the corresponding module is zero, it will cause a slow increase in the current on the load from zero, greatly slowing down the front speed of the pulse current. In addition, there are many factors affecting the inductive charging current, such as equivalent series resistance of the inductor and circuit and voltage drop of the IGBT switch. Because it is difficult to ensure the full consistency of parameters between modules, the overall cascade mode has inherent defects.

The Marx pulsed adder cascades in series, which is equivalent to forming a voltage stacking with more stages. The packet cascade structure is equivalent to a larger pulse current source, and the difference tolerance between modules is very high. Even if a module Marx pulsed adder cannot output voltage due to a fault, the inductor will still charge normally, which will only cause the charging current to decrease and will not cause problems such as a flat-top rise of the load current. Because the solid-state Marx pulsed adder and the inductor are separate, the number of series inductors can be adjusted separately to make the series inductance within a reasonable range. So, the packet cascade scheme is more suitable for the pulsed current source cascade system.

Using the packet cascade mode, we constitute a large pulse current source, which includes a nine-stage solid-state Marx pulsed adder and six high-voltage inductors (series inductance is approximately 3 H). The initial output voltage of the Marx stack is 170 kV. Figure 9 shows the photos of the cascade system of pulsed current source. The total series equivalent capacitance is ~70 nF. Each pulsed current module is shown in Figure 4.

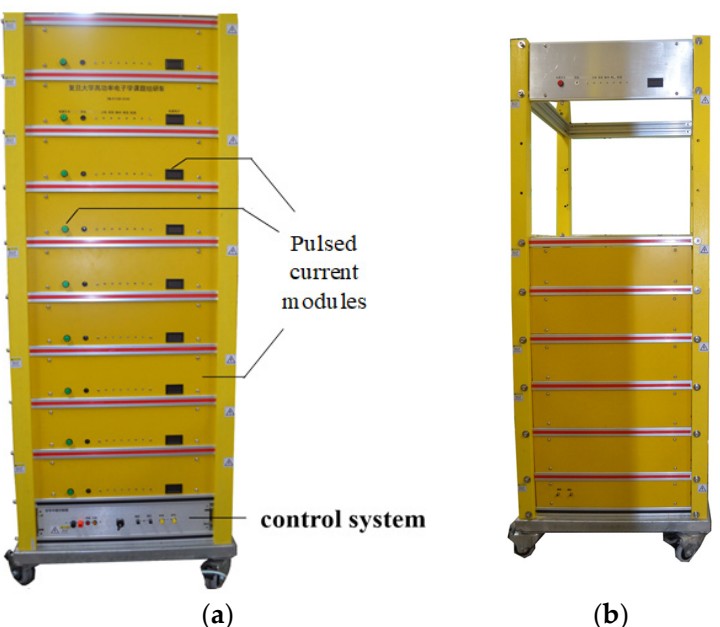

(**a**)                                        (**b**)

**Figure 9.** The photos of the cascade system of pulsed current source. (**a**) The cascaded solid-state Marx sources. (**b**) The cascaded high-voltage inductors.

Figure 10 shows the current waveform of the 10 kΩ resistance load driven by the cascade pulse current source. The maximum output voltage of the pulse current source is 155 kV. We used a PEARSON current monitor MODEL 110 A (with a bandwidth of

20 MHz) and high-voltage probe EP-150KP (manufactured by Nissin Pulse Electronics Co., Ltd., Yamazaki Noda-City Chiba, Japan. with a bandwidth of 50 MHz) to measure the pulsed current and voltage. Because the product of the load resistance and the inductive current is less than the maximum output voltage of the cascaded Marx pulsed adder, all the inductive current is released to the load. The time t corresponding to the curve in the figure is the time required for the solid-state Marx pulsed adder to charge the inductance. Compared with the pulse voltage adder, the pulse current source can obtain a more stable pulse current, especially in the pulse flat-top stage.

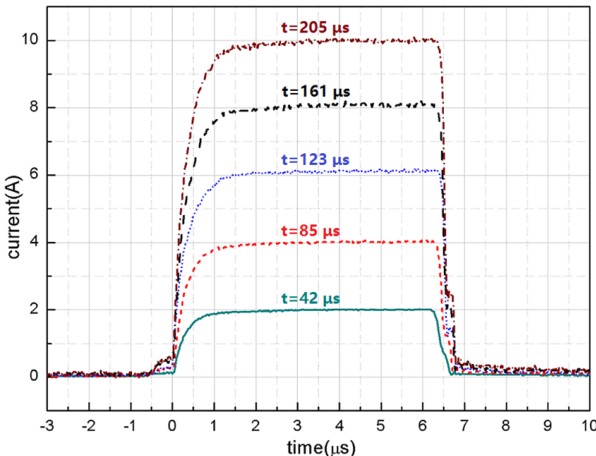

**Figure 10.** Output current of the cascade pulse current source.

There may be two conditions for the same pulsed current amplitude on the load: in the case of a current source without voltage limitation, all of the inductance current flows through the load; in the case of a current source with a voltage clamp, only part of the inductance current flows through the load, and the other current is fed back to the capacitors of the Marx pulsed adder. The following experimental results will indicate whether the current source is current-limiting, the driving resistance and load impedance and the current value of the inductive charge. In order to explore the change in the pulse rising edge when the pulsed current source generates pulsed current with lower amplitude, four sets of experiments were analyzed, as shown in Figure 11. Figure 11a,b show the small inductance current of 1.8 A, and the inductance current shall be released on the load. Figure 11a shows the 1.7 A current obtained on the 15 kΩ load, and Figure 11b shows the 1.5 A current obtained on the 60 kΩ load. Figure 11c shows the critical situation of the power supply; 7.5 A inductor current flows through the 20 kΩ load, and the voltage on the load is the current source voltage limit value. Figure 11d shows a heavy load and large current; the inductor current is 10 A, the 60 kΩ load receives 2.5 A current, and the voltage at the two ends of the load is limited.

The experimental results indicate that, when the inductor charging current is small, not all of the inductor current will be released to the load. In this case, the expected current amplitude should be obtained on the load, and the charging time of the inductor should be appropriately extended. When the inductor current is small, the front edge of the current pulse will become slow, while the actual load current rises along the theoretical rise point (that is, the time point when the inductor stops charging, and all IGBT switches are off; Figure 11a,b delay Δt reaches 10 μs. There is a weak current on the load during the delay period. This narrowed the current pulse width on the load, the set current pulse width is 22 μs, and the actual current pulse width is 12 μs. When the inductance current becomes larger, the rising speed of the current on the load is significantly faster, the current rise is closer to the theoretical current rising point, and the pulse width of the output current is in accord with the theoretical value.

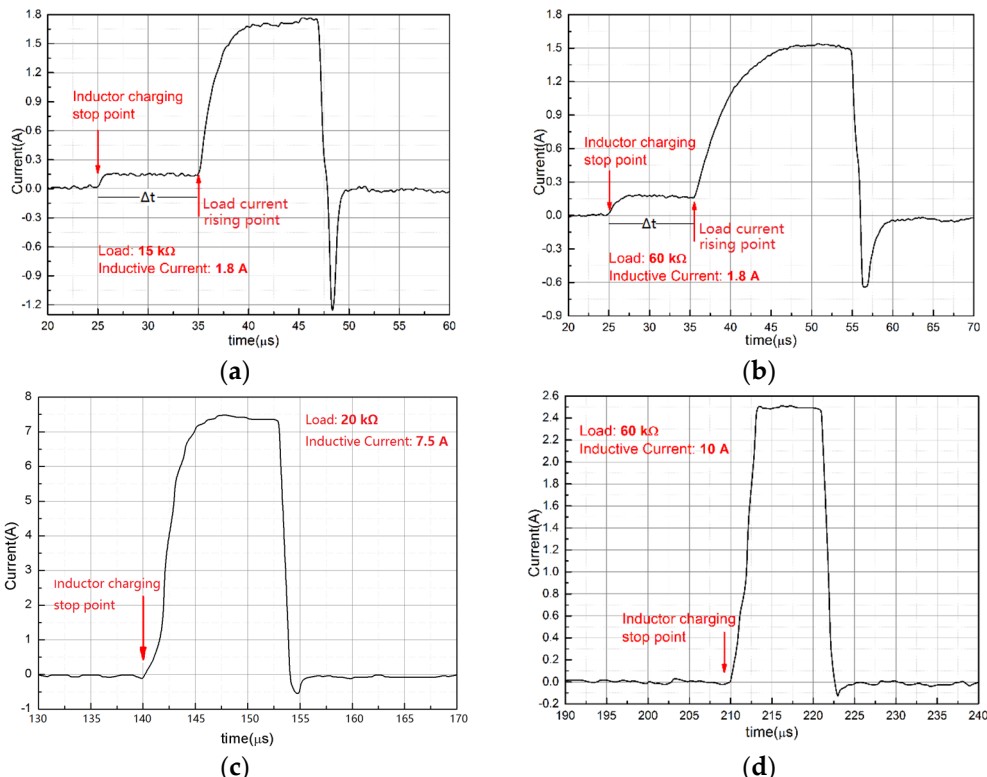

**Figure 11.** The output current pulse waveform for each condition of the cascade system. (**a**) 1.7 A @ 15 kΩ unlimited flow status. (**b**) 1.5 A @ 60 kΩ unlimited flow status. (**c**) 7.5 A @ 20 kΩ criticality. (**d**) 10 A @ 60 kΩ current-limiting status.

When the scale of the cascade system becomes larger, the influence of the distribution parameters becomes more significant, especially for the inhibition of the rate of pulse current rise. The distribution parameters need to be considered for the calculation, and some measures should be taken to reduce the distribution parameters for better current pulse waveforms.

## 5. Effects of the Distribution Parameters

After the experiment of the cascade system, it is found that the output waveform and the theoretical waveform are very different; the main difference is the current pulse rise time and pulse width. This is because the distribution parameters in the cascade system are relatively more significant. Figure 12 is the diagram of the pulse current source topology with the distribution parameters.

The capacitance $C_L$ is the equivalent stray capacitance of the inductor, the capacitance $C_{ss}$ is the equivalent stray capacitance of the high-voltage silicon stack, and the capacitor $C_s$ is the distributed capacitor between the IGBT switches and the ground, which is introduced by the driving line. In addition, because the solid-state pulsed current module is placed on the stack, there is a spatial capacitance between the different modules and the ground. The distributed capacitance of each solid-state pulsed current module is relatively scattered, so it is difficult to estimate the distributed capacitance value in the specific position. Therefore, the distributed capacitance of each solid-state pulsed current module is equivalent to the output end in parallel with the distributed capacitance of the inductor and merged into a parallel distributed capacitor $C_S$. All stray inductance in the loop is combined into an equivalent series inductance $L_{SC}$. The Marx pulsed adder can be equivalent by a series circuit of a capacitor, a resistor R and a switch $S_W$. Switch $S_C$ has the same function as the switches $S_{a-i}$ do when the switches $S_{b-i}$ are open in Figure 12. The equivalent circuit of the pulse current source is shown in Figure 13.

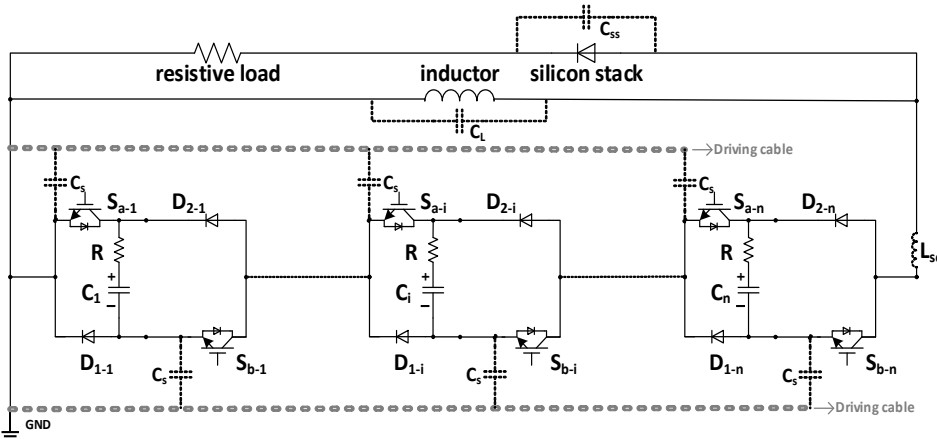

**Figure 12.** Diagram of the pulse current source topology with distribution parameters.

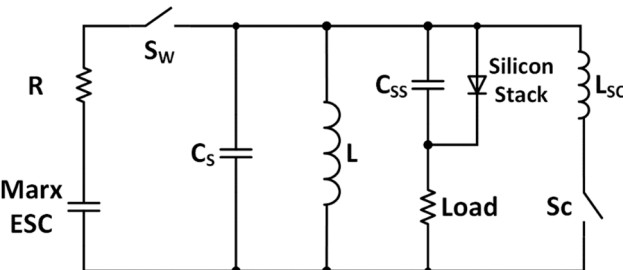

**Figure 13.** Equivalent circuit of the pulse current source including distribution parameters.

Figure 14 shows the current path diagram of the inductive charging stage, the blue dotted arrows represent the direction of current flow. the Marx output voltage is negative, and the distributed capacitors $C_S$ and $C_{SS}$ are charged to the negative voltage. Due to the influence of the high-voltage silicon stack distribution capacitor, the load will appear as a narrow pulse width negative current pulse during the falling time of the Marx pulsed adder output. The amplitude of the negative current on the load is determined by the falling time of the output pulsed and the distribution capacitance of the silicon stack. When the inductive charging is completed, the load negative current is zero.

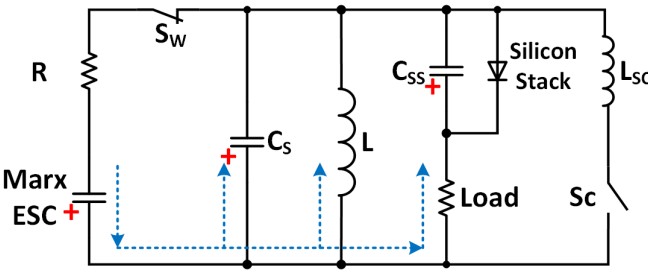

**Figure 14.** Diagram of the inductive charging stage.

When the Marx pulsed adder discharge switch $S_W$ is disconnected, the power supply enters the inductive discharge phase. The energy storage capacitor forms a series path through the diodes in the Marx pulsed adder (after the switch $S_W$ turns off, it can be replaced by a diode D), and the voltage is turned into a positive voltage. Since the distributed capacitors $C_S$ and $C_{SS}$ are charged to the negative voltage by the Marx pulsed adder, the distributed capacitor should be charged to zero first, as shown in Figure 15 in the front current path of the inductive discharge section. Part of the inductive current will be lost in charging the distributed capacitance, while a small part of the current is output to the load.

A high load current will not occur when the IGBT discharge switch is turned off, so the output current pulse width will decrease.

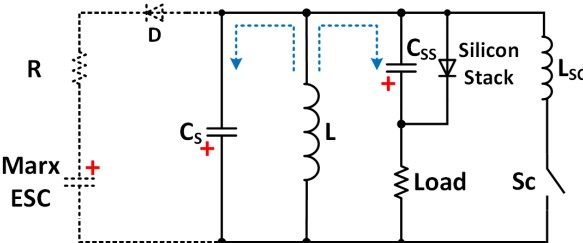

**Figure 15.** Current path before the inductive discharge phase.

When the distributed $C_S$ voltage is zero, the voltage on the $C_{SS}$ may still be negative, and the current still exists on the load. To simplify the analysis of the load current pulse front, it is assumed that the $C_{SS}$ voltage drops to zero earlier than the $C_S$, the voltage on the distributed capacitance is all zero, and the inductive discharge stage enters the second stage. The current path is shown in Figure 16. The inductive current charges the distributed capacitor $C_S$ (Path ①) and discharges the load. If the $C_S$ voltage reaches the capacitor limit value of the Marx pulsed adder, part of the inductive current will charge the energy storage capacitor back (Path ②).

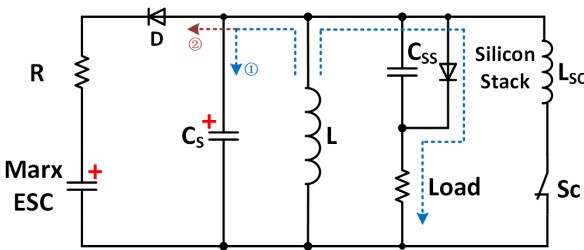

**Figure 16.** Current path after the inductive discharge phase.

Assuming the current through the load $R_{load}$ is $I_l$, the differential equation can be obtained:

$$\begin{cases} \ddot{I}_1 + \dfrac{\dot{I}_1}{C_s R_1} - \dfrac{I_1}{C_s L} = 0 \\ I_1(0) = 0, \dot{I}_1(0) = 0 \end{cases} \tag{1}$$

According to the formula, the inductance is set to 3 H, the $C_S$ is set to 40 pF, and the current rises along the waveform when different inductive currents drive the load with different resistance values, as shown in Figure 17. On the same load, the larger the inductive current is, the faster the load current rises; the smaller the load is, the faster the load current rises.

The switches $S_{b-i}$ are on at the same time, the inductive current will flow through the switches' loop, and the current on the load is rapidly reduced. At the same time, the distributed capacitor $C_S$ is also discharged through the continuous flow circuit. Due to the existence of the switches $S_{b-i}$ circuit distribution inductance, the load current will appear to reverse overshoot.

Figure 18 shows the six processes of a typical pulse current source. Process one occurs at the descending edge of the output voltage of the Marx pulsed adder. The current path is shown in Figure 14. The second process is the front section of the inductance to load discharge. The current path is shown in Figure 15. Since most of the inductance current preferentially charges the distributed capacitance, the current obtained on the load is very small. Process three is the following section, and the current path is shown in Figure 16. Process four is that the load current reaches the flat top; during the process, all

the inductance current flows through the load, or the voltage at the two ends of the load is limited by the capacitor voltage of the Marx pulsed adder. In Process five, the switches $S_{a-i}$ or $S_{b-i}$ turn on, load current cutoff process; the current path is shown in Figure 19. Due to the equivalent inductance of $L_{SC}$, the load current will show a negative overshoot. Process six is the transfer process of inductive energy storage; in the process, the switches $S_{a-i}$ and $S_{b-i}$ are disconnected, and the capacitor $C_i$ absorbs the energy of the inductor through Path ② shown in Figure 16.

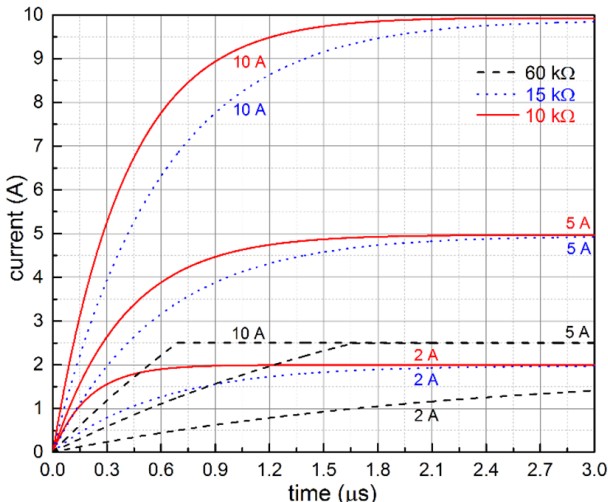

**Figure 17.** The current rise time with different loads.

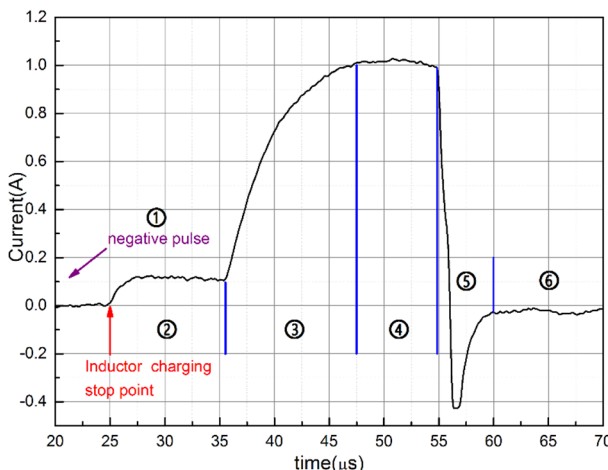

**Figure 18.** Typical current-pulse waveform process.

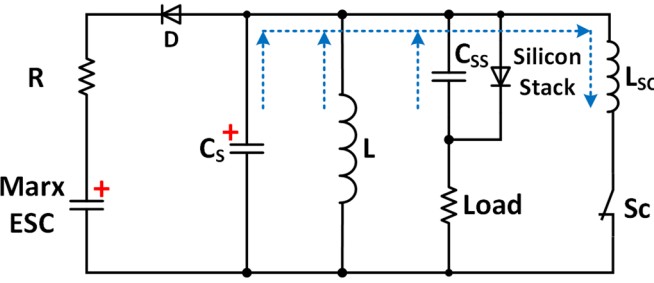

**Figure 19.** Current path during cutoff phase.

## 6. Conclusions

In this paper, a design of a high-voltage inductive pulsed current generator was expounded. One standard module manufactured is capable to drive loads with unfixed resistance changing in a certain range. Its circuit principle and control sequence were described. The circuit ensures the inductive current has a continuous path at all times, so the system can operate safely without the risk of overvoltage. The inductive pulsed current generator has some advantages, like the performance of current pulses. Composed of solid-state devices, it can be controlled by low-voltage logic devices through isolated drivers. Referring to the formulas given, it is easy to modulate a variety of current pulses with different parameters by changing the switching time.

There are two cascade methods for pulsed current sources: overall cascade and packet cascade. The overall cascade refers to taking a complete pulse current source as the basic unit of the cascade, forming a large power supply system through the connection between the pulse current sources. The packet cascade refers to the grouping of inductors and solid Marx stacks, and then the two groups are connected to form a larger power system. The overall cascade method is simple to connect, but it is extremely sensitive to the difference of the inductive charging current of different modules. Due to the engineering limitation, it is impossible to achieve complete consistency between modules, and the uneven inductive current cannot be avoided. In fact, the packet cascade method can be regarded as a magnified version of the pulse current source module. The series between the inductors avoids the inductor current imbalance, and the engineering realization is simple for the actual cascade system.

The experimental results of the cascade system show that, when the inductance current is small, the leading edge of the current pulse becomes slow, and the current pulse width is narrower than the theoretical calculation; when the inductance current becomes large, the current rise speed on the load is significantly faster, and the output current pulse width is closer to the theoretical value. In the cascade system, the distribution parameter of each unit of the pulse current source becomes larger, and the influence on the waveform becomes more obvious than that of the single-stage module. The distributed parameters in the cascade system will influence the output performance of the pulsed current generator, especially the distributed capacitance. Considering the distributed capacitance in the discharge processes, the current paths become more and complex. To reduce the influence of the distribution parameters, some measures could be taken, such as a reduction of the capacitance value by selecting devices with a lower distributed capacitance and more series devices and a Z-shaped circuit design to reduce discharge loop inductance.

In the future, it is proposed to add an independent controller for the pulsed current module, so each module can perform respectively by control command transmitted through optical fibers. Meanwhile, the performance of the cascade system would be optimized through feedback control for different output requirements. The flexibility of the system will be greatly enhanced.

**Author Contributions:** Conceptualization, W.D., J.Q. and K.L.; methodology, W.D. and J.Q.; software, L.C. and H.S.; validation, W.D.; formal analysis, L.C. and H.Z.; investigation, W.D. and J.Q.; writing—original draft preparation, W.D. and L.C.; writing—review and editing, J.Q., H.Z., K.L. and W.D.; supervision, J.Q. and H.Z.; project administration, J.Q. All authors have read and agreed to the published version of the manuscript.

**Funding:** This research received no external funding.

**Data Availability Statement:** All data that support the findings of this study are included within the article.

**Conflicts of Interest:** Author Haozheng Shi was employed by the company Shanghai Fudan Microelectronics Group Co., Ltd. The remaining authors declare that the research was conducted in the absence of any commercial or financial relationships that could be construed as a potential conflict of interest.

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
