# Peer review of "The Cascade of High-Voltage Pulsed Current Sources"

_electronics, doi:10.3390/electronics13050924_

Round 1

Reviewer 1 Report

Comments and Suggestions for Authors

Dear authors, thank you for submitting your article to Electronics. The content of the proposed manuscript does not bring significant scientific contribution in comparison to the existing literature. Some key issues found in the manuscript are detailed in the following.

1. The proposal should be supported by a detailed mathematical analysis. There is not a single equation in the manuscript. In my opinion, the results provided in [23] are not enough to represent the complete cascade system and its particularities. However, if the mathematical analysis detailed in [23] is enough, then the contribution of this new article is not significant.

2. Theoretical waveforms must be included to support the explanation provided in lines 83–99.

3. The circuit schematic depicted in Fig. 1 leads to the conclusion that the pulsed current source contains three cells instead of “n” cells. In order to achieve a proper generic representation of the circuit, please used dashed lines between the connection of cells “1”, “i”, and “n”. The same applies for other figures contained in the article.

4. The parameter “n” yields a conflict, since it is used to represent both the number of cells contained in each module of the cascade system and also the number of modules.

5. A better explanation of the delay period observed in the waveforms depicted in Fig. 12 (a) and (b) must be provided by the authors. The causes of this phenomenon must be clarified. Please provide the equivalent circuits of each interval highlighting the paths of current flow for a better visualization.

6. Please provide an explanation of the negative current peak observed in Fig. 12 (a) and (b).

7. Details on the condition of voltage clamp depicted in Fig. 12 (d) must be provided. A circuit containing highlighted paths of the current flow must be provided to demonstrate how voltage clamping is accomplished.

8. What is the value of the DC voltage source used in the experimental setup?

9. Based on the information provided in the manuscript and on the photographs depicted in Fig. 10, it is not clear how many cells are contained in each Marx source and modules in the cascade system.

10. Please provide photographs of the PCB of each Marx source as well.

11. It is known that inductances may vary due to manufacturing tolerances and aging, and therefore the current value might diverge from the desired value. As a result, a proper control system must be used to compensate for these variations. Please provide more details on the control system and how it acts to guarantee stable amplitude for the pulsed current.

12. What is the efficiency of the equipment used in the experimental tests?

Comments on the Quality of English Language

Some issues are found in the manuscript, detailed as follows:

1) “a higher output parameters” contains a grammar problem.

2) Unproper spacing is found in “( series”.

3) “when the pulse current source output a small current” ” contains a grammar problem.

Author Response

Dear Editors and Reviewers:

Thank you for your letter and for the reviewers´ comments concerning our manuscript entitled “The Cascade of High-Voltage Pulsed Current Sources” (ID: electronics-2851198). Those comments are all valuable and very helpful for revising and improving our paper, as well as the important guiding significance to our researches.

We agree with the reviewers that the details of the submitted manuscript should be improved and we have studied comments carefully and have made corrections which we hope meet with approval. Revised portions are marked in yellow in the paper. The main corrections in the paper and the response to the reviewer´s comments are as follows:

Reviewer 1

General recommendations:

1.Response to comment: (The proposal should be supported by a detailed mathematical analysis. There is not a single equation in the manuscript. In my opinion, the results provided in [23] are not enough to represent the complete cascade system and its particularities. However, if the mathematical analysis detailed in [23] is enough, then the contribution of this new article is not significant.)

Author response: Thank you very much for your valuable suggestion. I have added more detail analysis at the end of Section 5. In this manuscript, we focus on the feasibility of the cascade method and the effects of stray parameters in the cascade system, the main analysis is about the interpretation of the waveform. For details, please see the revised manuscript (lines 54-88, 300-390 ).

  1. Response to comment: ( Theoretical waveforms must be included to support the explanation provided in lines 83–99.)

Author response: Thank you very much for your valuable suggestion. I have added the related  content in the manuscript. For details, please see the revised manuscript (lines l35-l40) .

3.Response to comment: (The circuit schematic depicted in Fig. 1 leads to the conclusion that the pulsed current source contains three cells instead of “n” cells. In order to achieve a proper generic representation of the circuit, please used dashed lines between the connection of cells “1”, “i”, and “n”. The same applies for other figures contained in the article.)

Author response: Thank you very much for your valuable suggestion. I have improved the circuit schematic.

4.Response to comment: (The parameter “n” yields a conflict, since it is used to represent both the number of cells contained in each module of the cascade system and also the number of modules.)

Author response: Thank you very much for your valuable suggestion. I have used different parameters to avoid the conflict.

5.Response to comment: (A better explanation of the delay period observed in the waveforms depicted in Fig. 12 (a) and (b) must be provided by the authors. The causes of this phenomenon must be clarified. Please provide the equivalent circuits of each interval highlighting the paths of current flow for a better visualization.)

Author response: Thank you very much for your valuable suggestion. I have I have added more detail analysis at the end of Section 5 to explanation the distortion of the waveform.

6.Response to comment: (Please provide an explanation of the negative current peak observed in Fig. 12 (a) and (b).)

Author response: Thank you very much for your valuable suggestion. I have I have added more detail analysis at the end of Section 5 to explanation the distortion of the waveform.

7.Response to comment: ( Details on the condition of voltage clamp depicted in Fig. 12 (d) must be provided. A circuit containing highlighted paths of the current flow must be provided to demonstrate how voltage clamping is accomplished.)

Author response: Thank you very much for your valuable suggestion. I have I have added more detail analysis at the end of Section 5 to explanation the distortion of the waveform.

8.Response to comment: (What is the value of the DC voltage source used in the experimental setup?)

Author response: Because the pulsed current module has 26 stages, if we want to obtain the pulsed peak voltage with 20kV-24kV,  the value of the DC voltage source should be about 800~1000V.

9.Response to comment: (Based on the information provided in the manuscript and on the photographs depicted in Fig. 10, it is not clear how many cells are contained in each Marx source and modules in the cascade system.)

Author response:Thank you very much for your valuable suggestion. I have added the information in the manuscript. Please see the revised manuscript (lines 140-158).  

10.Response to comment: (Please provide photographs of the PCB of each Marx source as well.)

Author response:Thank you very much for your valuable suggestion. I have added the information in the manuscript. Please see the revised manuscript (lines 140-158).

11.Response to comment: (It is known that inductances may vary due to manufacturing tolerances and aging, and therefore the current value might diverge from the desired value. As a result, a proper control system must be used to compensate for these variations. Please provide more details on the control system and how it acts to guarantee stable amplitude for the pulsed current.)

Author response: Thanks for your attention, it is a good question. Actually, there are two meanings about the stable of pulsed current, one is the amplitude of each pulsed current, the other is the amplitude during the current pulse. For the former, we usually use the closed-loop feedback control method, and some methods are also mentioned in the manuscript. Our circuit scheme is actually designed for the latter, mainly using the energy storage characteristics of the inductor to realize that the output current remains almost the same with the time-varying load during the pulse,for example,the electric breakdown on the load.

12.Response to comment: (What is the efficiency of the equipment used in the experimental tests?)

Author response:Because the main loss occurs in the transfer process of the inductive energy to the capacitors, the efficiency depends on the repetition rate of the experimental equipment. The actual pulsed frequency we used is very low, so the corresponding efficiency is not very high.

English recommendations:

1.Response to comment: ( “a higher output parameters” contains a grammar problem.)

Author response: Thanks for pointing out the problem, we are sorry for the stupid mistake we made. We have modified.

2.Response to comment: (Unproper spacing is found in “( series”.)

Author response: Thanks for pointing out the problem, we are sorry for the stupid mistake we made. We have modified.

3.Response to comment: ( “when the pulse current source output a small current” ” contains a grammar problem.)

Author response: Thanks for pointing out the problem, we are sorry for the stupid mistake we made. We have modified.

We tried our best to improve the manuscript and made some changes to the manuscript. These changes will not influence the content and framework of the paper. We appreciate for Editors and Reviewers warm work earnestly and hope that the correction will meet with approval. Once again, thank all of you very much for your comments and suggestions.

Sincerely,

Weigang Dong

Fudan University

Reviewer 2 Report

Comments and Suggestions for Authors

This manuscript proposes a novel design for a pulsed current source, utilizing a solid-state Marx pulsed adder as the primary power source and an inductor for energy storage. The system generates high-voltage square current pulses by charging the inductor with the Marx pulsed adder. The amplitude of the pulsed current remains unaffected by the load impedance within a certain range. The pulsed current source is designed as a standard module, and multiple modules can be cascaded to produce current pulses with higher voltage. The feasibility of the cascade system is validated through experiments, where nine modules generate adjustable current pulses of 10 A on a 15 Ω resistor. The cascade system, operating even in the presence of insulation breakdown on the load, limits the output pulsed current to prevent damage to load electrodes. The manuscript also discusses two cascade methods, overall cascade, and packet cascade, with the latter being more practical for achieving consistent inductive charging current between modules. Experimental results reveal insights into the effects of inductance current on the pulse characteristics in the cascade system, emphasizing the importance of analyzing and optimizing distribution parameters for waveform optimization. The topic is interesting, and the manuscript is well prepared. However, the reviewer has some concerns regarding the paper's contributions and application in this manuscript, so this manuscript is not ready for publication in its current form.

Here are some questions and comments:

ü  In the introduction section, there is a notable opportunity for improvement in the literature review's depth and comprehensiveness. The authors could enhance the introduction by incorporating recent published research developments, offering a more detailed exploration of the current state of the field. It would be beneficial to provide a comprehensive overview of the historical development of the topic, offering readers a contextual understanding of the evolution of research in this domain.

ü  It is essential that the parameters used in the manuscript are clearly defined. The authors should consistently address this issue throughout the manuscript. Additionally, ensure that all parameter values are included when presenting numerical results.

ü  The manuscript is marred by numerous grammatical mistakes that permeate its entirety. A comprehensive review and thorough editing are strongly recommended to rectify these errors, ensuring the manuscript attains a higher standard of linguistic precision and clarity.

ü  The authors should explain the advantages of their proposed topology over other existing structures. They need to provide a comparative analysis with other topologies to demonstrate its efficiency.

ü  In addition to comparing their cascaded pulsed current source structure with other recently published pulsed current source structures, the authors should clearly highlight the conceptual advantages of their cascaded topology compared to the recently published advanced structures. Emphasize the novelty of their manuscript, which revolves around their proposed framework.

ü  The quality of the figures in the paper, especially figures 2, 6, 9 and 12, is notably low. The authors should work on enhancing the figure quality throughout the manuscript.

ü  Provide more detailed explanations for the results presented in figures 8, 11 and 12.

ü  The conclusion should present more detailed results to wrap up the manuscript effectively.

Comments on the Quality of English Language

ü  The manuscript is marred by numerous grammatical mistakes that permeate its entirety. A comprehensive review and thorough editing are strongly recommended to rectify these errors, ensuring the manuscript attains a higher standard of linguistic precision and clarity.

Author Response

Dear Editors and Reviewers:

Thank you for your letter and for the reviewers´ comments concerning our manuscript entitled “The Cascade of High-Voltage Pulsed Current Sources” (ID: electronics-2851198). Those comments are all valuable and very helpful for revising and improving our paper, as well as the important guiding significance to our researches.

We agree with the reviewers that the details of the submitted manuscript should be improved and we have studied comments carefully and have made corrections which we hope meet with approval. Revised portions are marked in yellow in the paper. The main corrections in the paper and the response to the reviewer´s comments are as follows:

Reviewer 2

General recommendations:

1.Response to comment: (In the introduction section, there is a notable opportunity for improvement in the literature review's depth and comprehensiveness. The authors could enhance the introduction by incorporating recent published research developments, offering a more detailed exploration of the current state of the field. It would be beneficial to provide a comprehensive overview of the historical development of the topic, offering readers a contextual understanding of the evolution of research in this domain.)

Author response: Thank you very much for your valuable suggestion. I have provide more detailed exploration of the current state of the field. Please see the revised manuscript (lines 54-87).

2.Response to comment: (It is essential that the parameters used in the manuscript are clearly defined. The authors should consistently address this issue throughout the manuscript. Additionally, ensure that all parameter values are included when presenting numerical results.)

Author response: Thank you very much for your valuable suggestion. We have modified.

3.Response to comment: (The manuscript is marred by numerous grammatical mistakes that permeate its entirety. A comprehensive review and thorough editing are strongly recommended to rectify these errors, ensuring the manuscript attains a higher standard of linguistic precision and clarity.)

Author response: Thank you very much for your valuable suggestion. We are sorry for the stupid mistake we made.

4.Response to comment: (The authors should explain the advantages of their proposed topology over other existing structures. They need to provide a comparative analysis with other topologies to demonstrate its efficiency.)

Author response: Thank you very much for your valuable suggestion. I have provide some information in the manuscript. Please see the revised manuscript (lines 54-88).

5.Response to comment: (In addition to comparing their cascaded pulsed current source structure with other recently published pulsed current source structures, the authors should clearly highlight the conceptual advantages of their cascaded topology compared to the recently published advanced structures. Emphasize the novelty of their manuscript, which revolves around their proposed framework.)

Author response: Thank you very much for your valuable suggestion. I have provide the  information in the first part of the manuscript.

6.Response to comment: (The quality of the figures in the paper, especially figures 2, 6, 9 and 12, is notably low. The authors should work on enhancing the figure quality throughout the manuscript.)

Author response: Thank you very much for your valuable suggestion. I have improved the figures. In order to better highlight the focus, Figure 6 and 9 have been removed.

7.Response to comment: (Provide more detailed explanations for the results presented in figures 8, 11 and 12.)

Author response: Thank you very much for your valuable suggestion. I have provide more explanation in the manuscript. Please see the revised manuscript (lines 300-390).

8.Response to comment: (The conclusion should present more detailed results to wrap up the manuscript effectively.)

Author response: Thank you very much for your valuable suggestion. I have expanded the content of the conclusion.

English recommendations:

1.Response to comment: (The manuscript is marred by numerous grammatical mistakes that permeate its entirety. A comprehensive review and thorough editing are strongly recommended to rectify these errors, ensuring the manuscript attains a higher standard of linguistic precision and clarity.)

Author response: Thanks for pointing out the problem, we are sorry for the stupid mistake we made.

We tried our best to improve the manuscript and made some changes to the manuscript. These changes will not influence the content and framework of the paper. We appreciate for Editors and Reviewers warm work earnestly and hope that the correction will meet with approval. Once again, thank all of you very much for your comments and suggestions.

Sincerely,

Weigang Dong

Fudan University

Reviewer 3 Report

Comments and Suggestions for Authors

The paper has experimental validation. However, some important modifications should be done in the manuscript:

1) Add figures to better explain the operating processes of the pulse current module in Figure 1.

2) Explain the main contributions of the proposed pulse current module respect to other approaches in literature.

3) It is indicated in the abstract that the amplitude of the pulse current is unaffected by the load impedance within a certain range. However, neither simulation nor experimental results prove that affirmation. Please, explain this affirmation and add some results that proves it.

4)  Propose some solutions for the problem in the overall cascade method due to difference of the inductive charging current of different modules. 

5) Add more references about pulsed adders publised in the last 5 years, e.g.:
Jin, Y.; Cheng, L. An Inductive Isolation-Based 10 kV Modular Solid Boost-Marx Pulse Generator. Electronics 2023, 12, 1586. https://doi.org/10.3390/electronics12071586 

Comments on the Quality of English Language

Some minor mistakes should be corrected.

Author Response

Dear Editors and Reviewers:

Thank you for your letter and for the reviewers´ comments concerning our manuscript entitled “The Cascade of High-Voltage Pulsed Current Sources” (ID: electronics-2851198). Those comments are all valuable and very helpful for revising and improving our paper, as well as the important guiding significance to our researches.

We agree with the reviewers that the details of the submitted manuscript should be improved and we have studied comments carefully and have made corrections which we hope meet with approval. Revised portions are marked in yellow in the paper. The main corrections in the paper and the response to the reviewer´s comments are as follows:

General recommendations:

1.Response to comment: ( Add figures to better explain the operating processes of the pulse current module in Figure 1.)

Author response: Thank you very much for your valuable suggestion. Please see the revised manuscript (lines l35-l54). I also provide Reference [28] for more detailed instructions.

2.Response to comment: (Explain the main contributions of the proposed pulse current module respect to other approaches in literature.)

Author response: Thank you very much for your valuable suggestion. I have provide some information in the manuscript. Please see the revised manuscript (lines 54-88).

3.Response to comment: (It is indicated in the abstract that the amplitude of the pulse current is unaffected by the load impedance within a certain range. However, neither simulation nor experimental results prove that affirmation. Please, explain this affirmation and add some results that proves it.)

Author response: Thank you very much for your valuable suggestion. In Section 5, Figure 17 shows the constant-current characteristic. Actually, there are two meanings about the stable of pulsed current, one is the amplitude of each pulsed current, the other is the amplitude during the current pulse. For the former, we usually use the closed-loop feedback control method, and some methods are also mentioned in the manuscript. Our circuit scheme is actually designed for the latter, mainly using the energy storage characteristics of the inductor to realize that the output current remains almost the same with the time-varying load during the pulse,for example,the electric breakdown on the load.

4.Response to comment: ( Propose some solutions for the problem in the overall cascade method due to difference of the inductive charging current of different modules.)

Author response: Thank you very much for your valuable suggestion. In our opinion, overall cascade method includes many uncertainties, and although this method makes the power system more compact, it is not recommended for the reliable performance. Since each module is individually controlled, if the charging feedback function of the module can be added, the reliable operation of the overall cascade can be achieved.

5.Response to comment: (Add more references about pulsed adders publised in the last 5 years, e.g.:Jin, Y.; Cheng, L. An Inductive Isolation-Based 10 kV Modular Solid Boost-Marx Pulse Generator. Electronics 2023, 12, 1586. https://doi.org/10.3390/electronics12071586.)

Author response: Thank you very much for your valuable suggestion. 

English recommendations:

1.Response to comment: (Some minor mistakes should be corrected.)

Author response: Thanks for pointing out the problem, we are sorry for the stupid mistake we made.

We tried our best to improve the manuscript and made some changes to the manuscript. These changes will not influence the content and framework of the paper. We appreciate for Editors and Reviewers warm work earnestly and hope that the correction will meet with approval. Once again, thank all of you very much for your comments and suggestions.

Sincerely,

Weigang Dong

Fudan University

Reviewer 4 Report

Comments and Suggestions for Authors

-Literature references should be up to date.
-Figure 1 should be in better quality.
-How was the measurement performed on a real source?
-What instruments were used for the measurements?
-It would be appropriate to compare the results with similar works in the literature.
-What is the behavior of the source with inductive and capacitive load?

Author Response

Dear Editors and Reviewers:

Thank you for your letter and for the reviewers´ comments concerning our manuscript entitled “The Cascade of High-Voltage Pulsed Current Sources” (ID: electronics-2851198). Those comments are all valuable and very helpful for revising and improving our paper, as well as the important guiding significance to our researches.

We agree with the reviewers that the details of the submitted manuscript should be improved and we have studied comments carefully and have made corrections which we hope meet with approval. Revised portions are marked in yellow in the paper. The main corrections in the paper and the response to the reviewer´s comments are as follows:

General recommendations:

1.Response to comment: ( Literature references should be up to date.)

Author response: Thank you very much for your valuable suggestion. I have updated the references.  

2.Response to comment: (Figure 1 should be in better quality.)

Author response: Thank you very much for your valuable suggestion. I have improved the figure. 

3.Response to comment: (How was the measurement performed on a real source?)

Author response: Just use a current monitor and a high voltage probe.

4.Response to comment: ( What instruments were used for the measurements?)

Author response: Please see the revised manuscript (lines 250-252). 

5.Response to comment: (It would be appropriate to compare the results with similar works in the literature.)

Author response: Thank you very much for your valuable suggestion. I have provide some information in the manuscript. Please see the revised manuscript (lines 54-88).

6.Response to comment: (What is the behavior of the source with inductive and capacitive load?)

Author response: In Section 5, the equivalent circuit could answer this question. In my opinion, there would be a peak voltage on the inductive load when the current transfer to the load from the inductor. Meanwhile, the capacitive load would slow down the edge of the current pulses.

We tried our best to improve the manuscript and made some changes to the manuscript. These changes will not influence the content and framework of the paper. We appreciate for Editors and Reviewers warm work earnestly and hope that the correction will meet with approval. Once again, thank all of you very much for your comments and suggestions.

Sincerely,

Weigang Dong

Fudan University

Round 2

Reviewer 1 Report

Comments and Suggestions for Authors

Dear authors, thank you for submitting your revised article to Electronics. The manuscript has been improved and my comments addressed properly. Congratulations!

Author Response

    Thank you very much, your comments are all valuable and very helpful for revising and improving our paper, as well as the important guiding significance to our researches.

Reviewer 2 Report

Comments and Suggestions for Authors

The revised paper has been improved. However, I still have some minor comments on this manuscript as follows:

  1. The format of references should be improved according to the Journal style.
  2. In the conclusion section, the authors are suggested to add some potential future works about your current work.
  3. Please check the paper's writing mistakes carefully.
Comments on the Quality of English Language
  1. Please check the paper's writing mistakes carefully.

Author Response

Thank you very much, your comments are all valuable and very helpful for revising and improving our paper, as well as the important guiding significance to our researches.

  1. The format of references has been improved according to the Journal style.
  2. "In the future, it is proposed to add an independent controller for the pulsed current module, so each module can perform respectively by control command transmitted through optical fibers. Meanwhile,the performance of the cascade system would be optimized through feedback control for different output requirements, the flexibility of the system will be greatly enhanced." the words have been added in the conclusion.

  3. We have checked the paper's writing mistakes carefully. Thanks for your suggestion.

Reviewer 3 Report

Comments and Suggestions for Authors

My questions were solved.

Wirte the reference into MDPI style. 

Comments on the Quality of English Language

Good enough for publication

Author Response

Thank you very much. Those comments are all valuable and very helpful for revising and improving our paper, as well as the important guiding significance to our researches.

We have improved the references according to MDPI style.